# Depthwise Separable Convolutions for Neural Machine Translation

**Łukasz Kaiser**[*]
Google Brain
`lukaszkaiser@google.com`

**Aidan N. Gomez**[*][†]
University of Toronto
`aidan@cs.toronto.edu`

**François Chollet**[*]
Google Brain
`fchollet@google.com`

## Abstract

Depthwise separable convolutions reduce the number of parameters and computation used in convolutional operations while increasing representational efficiency. They have been shown to be successful in image classification models, both in obtaining better models than previously possible for a given parameter count (the Xception architecture) and considerably reducing the number of parameters required to perform at a given level (the MobileNets family of architectures). Recently, convolutional sequence-to-sequence networks have been applied to machine translation tasks with good results. In this work, we study how depthwise separable convolutions can be applied to neural machine translation. We introduce a new architecture inspired by Xception and ByteNet, called SliceNet, which enables a significant reduction of the parameter count and amount of computation needed to obtain results like ByteNet, and, with a similar parameter count, achieves better results. In addition to showing that depthwise separable convolutions perform well for machine translation, we investigate the architectural changes that they enable: we observe that thanks to depthwise separability, we can increase the length of convolution windows, removing the need for filter dilation. We also introduce a new "super-separable" convolution operation that further reduces the number of parameters and computational cost of the models.

## 1 Introduction

In recent years, sequence-to-sequence recurrent neural networks (RNNs) with long short-term memory (LSTM) cells (Hochreiter & Schmidhuber, 1997) have proven successful at many natural language processing (NLP) tasks, including machine translation (Sutskever et al., 2014; Bahdanau et al., 2014; Cho et al., 2014b). In fact, the results they yielded have been so good that the gap between human translations and machine translations has narrowed significantly (Wu et al., 2016) and LSTM-based recurrent neural networks have become standard in natural language processing.

Even more recently, auto-regressive convolutional models have proven highly effective when applied to audio (van den Oord et al., 2016a), image (van den Oord et al., 2016b) and text generation (Kalchbrenner et al., 2016). Their success on sequence data in particular rivals or surpasses that of previous recurrent models (Kalchbrenner et al., 2016; Gehring et al., 2017). Convolutions provide the means for efficient non-local referencing across time without the need for the fully sequential processing of RNNs. However, a major critique of such models is their computational complexity and large parameter count. These are the principal concerns addressed within this work: inspired by the efficiency of depthwise separable convolutions demonstrated in the domain of vision, in particular the Xception architecture (Chollet, 2016) and MobileNets (Howard et al., 2017), we generalize these techniques and apply them to the language domain, with great success.

---

[*]All authors contributed equally and are ordered randomly.

[†]Work performed while at Google Brain.
Code available at `https://github.com/tensorflow/tensor2tensor`

## 2    OUR CONTRIBUTION

We present a new convolutional sequence-to-sequence architecture, dubbed SliceNet, and apply it to machine translation tasks, achieving results that surpass all previous reported experiments except for the recent Transformer model (Vaswani et al., 2017). Our architecture features two key ideas:

- Inspired by the Xception network (Chollet, 2016), our model is a stack of depthwise separable convolution layers with residual connections. Such an architecture has been previously shown to perform well for image classification. We also experimented with using grouped convolutions (or "sub-separable convolutions") and add even more separation with our new super-separable convolutions.
- We do away with filter dilation in our architecture, after exploring the trade-off between filter dilation and larger convolution windows. Filter dilation was previously a key component of successful 1D convolutional architectures for sequence-to-sequence tasks, such as ByteNet (Kalchbrenner et al., 2016) and WaveNet (van den Oord et al., 2016a), but we obtain better results without dilation thanks to separability.

### 2.1    SEPARABLE CONVOLUTIONS AND GROUPED CONVOLUTIONS

The depthwise separable convolution operation can be understood as related to both grouped convolutions and the "inception modules" used by the Inception family of convolutional network architectures, a connection explored in Xception (Chollet, 2016). It consists of a depthwise convolution, i.e. a spatial convolution performed independently over every channel of an input, followed by a pointwise convolution, i.e. a regular convolution with 1x1 windows, projecting the channels computed by the depthwise convolution onto a new channel space. The depthwise separable convolution operation should not be confused with spatially separable convolutions, which are also often called "separable convolutions" in the image processing community.

Their mathematical formulation is as follow (we use $\odot$ to denote the element-wise product):

$$\text{Conv}(W, y)_{(i,j)} = \sum_{k,l,m}^{K,L,M} W_{(k,l,m)} \cdot y_{(i+k,j+l,m)}$$

$$\text{PointwiseConv}(W, y)_{(i,j)} = \sum_{m}^{M} W_m \cdot y_{(i,j,m)}$$

$$\text{DepthwiseConv}(W, y)_{(i,j)} = \sum_{k,l}^{K,L} W_{(k,l)} \odot y_{(i+k,j+l)}$$

$$\text{SepConv}(W_p, W_d, y)_{(i,j)} = \text{PointwiseConv}_{(i,j)}(W_p, \text{DepthwiseConv}_{(i,j)}(W_d, y))$$

Thus, the fundamental idea behind depthwise separable convolutions is to replace the feature learning operated by regular convolutions over a joint "space-cross-channels realm" into two simpler steps, a spatial feature learning step, and a channel combination step. This is a powerful simplification under the oft-verified assumption that the 2D or 3D inputs that convolutions operate on will feature both fairly independent channels and highly correlated spatial locations.

A deep neural network forms a chain of differentiable feature learning modules, structured as a discrete set of units, each trained to learn a particular feature. These units are subsequently composed and combined, gradually learning higher and higher levels of feature abstraction with increasing depth. Of significance is the availability of dedicated feature pathways that are merged together later in the network; this is one property enabled by depthwise separable convolutions, which define independent feature pathways that are later merged. In contrast, regular convolutional layers break this creed by learning filters that must simultaneously perform the extraction of spatial features and their merger into channel dimensions; an inefficient and ineffective use of parameters.

Grouped convolutions (or "sub-separable convolutions") are an intermediary step between regular convolutions and depthwise separable convolutions. They consist in splitting the channels of an input into several non-overlapping segments (or "groups"), performing a regular spatial convolution over each segment independently, then concatenating the resulting feature maps along the channel axis.

| Convolution type | Parameters and approximate floating point operations per position |
|---|---|
| Non-separable | $k \cdot c^2$ |
| Fully-separable | $k \cdot c + c^2$ |
| $g$-Sub-separable | $k \cdot \frac{c^2}{g} + c^2$ |
| $g$-Super-separable | $k \cdot c + \frac{c^2}{g}$ |

Table 1: Parameter count comparison across convolution types.

Depthwise separable convolutions have been previously shown in Xception (Chollet, 2016) to allow for image classification models that outperform similar architectures with the same number of parameters, by making more efficient use of the parameters available for representation learning. In MobileNets (Howard et al., 2017), depthwise separable convolutions allowed to create very small image classification models (e.g. 4.2M parameters for 1.0 MobileNet-224) that retained much of the capabilities of architectures that are far larger (e.g. 138M parameters for VGG16), again, by making more efficient use of parameters.

The theoretical justifications for replacing regular convolution with depthwise separable convolution, as well as the strong gains achieved in practice by such architectures, are a significant motivation for applying them to 1D sequence-to-sequence models.

The key gains from separability can be seen when comparing the number of parameters (which in this case corresponds to the computational cost too) of separable convolutions, group convolutions, and regular convolutions. Assume we have $c$ channels and filters (often $c = 1000$ or more) and a receptive field of size $k$ (often $k = 3$ but we will use $k$ upto 63). The number of parameters for a regular convolution, separable convolution, and group convolution with $g$ groups is:

$$k \cdot c^2 \qquad\qquad k \cdot c + c^2 \qquad\qquad k \cdot \frac{c^2}{g} + c^2.$$

## 2.2 SUPER-SEPARABLE CONVOLUTIONS

As can be seen above, the size (and cost) of a separable convolution with $c$ channels and a receptive field of size $k$ is $k \cdot c + c^2$. When $k$ is small compared to $c$ (as is usuallty the case) the term $c^2$ dominates, which raises the question how it could be reduced. We use the idea from group convolutions and the recent separable-LSTM paper (Kuchaiev & Ginsburg, 2017) to further reduce this size by factoring the final $1 \times 1$ convolution, and we call the result a *super-separable* convolution.

We define a super-separable convolution (denoted $SuperSC$) with $g$ groups as follows. Applied to a tensor $x$, we first split $x$ on the depth dimension into $g$ groups, then apply a separable convolution to each group separately, and then concatenate the results on the depth dimension.

$$\text{SuperSC}_g(\overline{W_p}, \overline{W_d}, x) = \text{concat}_{\text{depth}}(\text{SepConv}(W_p^1, W_d^1, x^1), \dots, \text{SepConv}(W_p^g, W_d^g, x^g)),$$

where $x^1, \dots, x^g$ is $x$ split on the depth axis and $W_p^i, W_d^i$ for $i = 1, \dots, g$ are the parameters of each separable convolution. Since each $W_d^i$ is of size $k \cdot \frac{c}{g}$ and each $W_p^i$ is of size $\frac{c^2}{g^2}$, the final size of a super-separable convolution is $k \cdot c + \frac{c^2}{g}$. Parameter counts (or computational budget per position) for all convolution types are summarized in Table 1.

Note that a super-separable convolution doesn't allow channels in separate groups to exchange information. To avoid making a bottleneck of this kind, we use stack super-separable convolutions in layer with co-prime $g$. In particular, in our experiments we always alternate $g = 2$ and $g = 3$.

## 2.3 FILTER DILATION AND CONVOLUTION WINDOW SIZE

Filter dilation, as introduced in (Yu & Koltun, 2015), is a technique for aggregating multiscale information across considerably larger receptive fields in convolution operations, while avoiding an explosion in parameter count for the convolution kernels. It has been presented in (Kalchbrenner et al.,

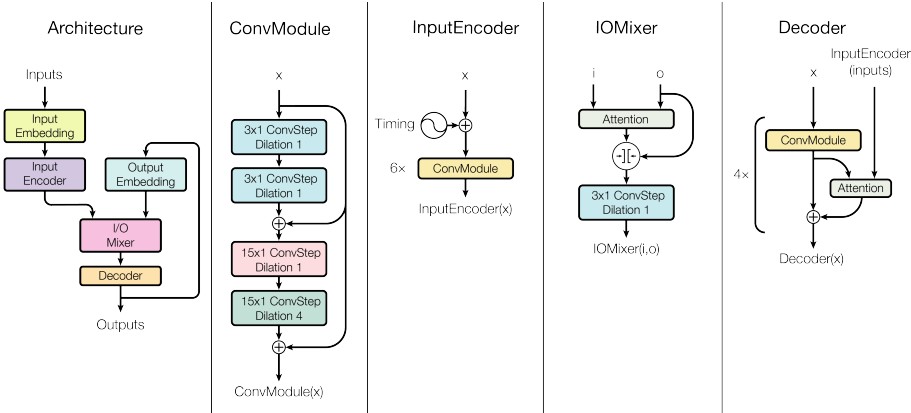

Figure 1: Summary of the SliceNet architecture. See text for a detailed explanation and equations.

2016) and (van den Oord et al., 2016a) as a key component of convolutional sequence-to-sequence autoregressive architectures.

When dilated convolution layers are stacked such that consecutive layers' dilation values have common divisors, an issue similar to the checkerboard artifacts in deconvolutions (Odena et al., 2016) appears. Uneven filter coverage results in dead zones where filter coverage is reduced (as displayed in the plaid-like appearance of Figure 1 in (Yu & Koltun, 2015)). Choosing dilation factors that are co-prime can indeed offer some relief from these artifacts, however, it would be preferable to do away with the necessity for dilation entirely.

The purpose of filter dilation is to increase the receptive field of the convolution operation, i.e. the spatial extent from which feature information can be gathered, at a reasonable computational cost. A similar effect would be achieved by simply using larger convolution windows. Besides, the use of larger windows would avoid an important shortcoming of filter dilation, unequal convolutional coverage of the input space. Notably, the use of depthwise separable convolutions in our network in place of regular convolutions makes each convolution operation significantly cheaper (we are able to cut the number of non-embedding model parameters by half), thus lifting the computational and memory limitations that guided the development of filter dilation in the first place.

In our experiments, we explore the trade-off between using lower dilation rates and increasing the size of the convolution windows for our depthwise separable convolution layers. In contrast to the conclusions drawn in WaveNet and ByteNet, we find that the computational savings brought on by depthwise separable convolutions allow us to do away with dilation entirely. In fact, we observe no benefits of dilations: our best models feature larger filters and no dilation (see Table 2). A comparison of the parameter count for different convolution operations is found in Table 1.

## 3 SLICENET ARCHITECTURE

Here we present the model we use for our experiments, called SliceNet in reference to the way separable convolutions operate on channel-wise slices of their inputs. Our model follows the convolutional autoregressive structure introduced by ByteNet (Kalchbrenner et al., 2016), WaveNet (van den Oord et al., 2016a) and PixelCNN (van den Oord et al., 2016b). Inputs and outputs are embedded into the same feature depth, encoded by two separate sub-networks and concatenated before being fed into a decoder that autoregressively generates each element of the output. At each step, the autoregressive decoder produces a new output prediction given the encoded inputs and the encoding of the existing predicted outputs. The encoders and the decoder (described in Section 3.3) are constructed from stacks of convolutional modules (described in Section 3.1) and attention (described in Section 3.2) is used to allow the decoder to get information from the encoder.

### 3.1 Convolutional Modules

To perform local computation, we use modules of convolutions with ReLU non-linearities and layer normalization. A module of convolutions gets as input a tensor of shape [sequence length, feature channels] and returns a tensor of the same shape. Each step in our module consist of three components: a ReLU activation of the inputs, followed by a depthwise separable convolution, followed by layer normalization. Layer normalization (Ba et al., 2016) acts over the $h$ hidden units of the layer below, computing layer-wise statistics and normalizing accordingly. These normalized units are then scaled and shifted by scalar learned parameters $G$ and $B$ respectively, producing the final units to be activated by a non-linearity:

$$\text{LN}(x) = \frac{G}{\sigma(x)}(x - \mu(x)) + B \qquad \sigma(x) = \sqrt{\frac{1}{h}\sum_i^h (x_i - \mu(x))^2} \qquad \mu(x) = \frac{1}{h}\sum_i^h x_i,$$

where the sum are taken only over the last (depth) dimension of $x$, and $G$ and $B$ are learned scalars.

A complete convolution step with kernel size $K$ and dilation $D$ is defined as:

$$\text{ConvStep}_{k=K,d=D}(x) = \text{LN}(\text{SepConv}(W_p, W_d, \text{ReLU}(x))),$$

where $W_p$ and $W_d$ are fresh sets of trainable weights that we omit from the notation for clarity. The convolutional steps are composed into modules by stacking them and adding residual connections as depicted in Figure 1. We use stacks of four convolutional steps with two skip-connections between the stack input and the outputs of the second and fourth convolutional steps:

$$\text{hidden}_1(x) = \text{ConvStep}_{k=3,d=1}(x)$$

$$\text{hidden}_2(x) = x + \text{ConvStep}_{k=3,d=1}(\text{hidden}_1(x))$$

$$\text{hidden}_3(x) = \text{ConvStep}_{k=15,d=1}(\text{hidden}_2(x))$$

$$\text{hidden}_4(x) = x + \text{ConvStep}_{k=15,d=4}(\text{hidden}_3(x))$$

$$\text{ConvModule}(x) = \begin{cases} \text{dropout}(\text{hidden}_4(x), 0.5) & \text{during training} \\ \text{hidden}_4(x) & \text{otherwise} \end{cases}$$

Figure 2: The ConvModule architecture described in Section 3.1. We vary the convolution sizes and dilations; see Section 3 for details on the architecture and Section 5 for the variations we study.

ConvModules are used in stacks in our module, the output of the last feeding into the next. We denote a stack with $n$ modules by $\text{ConvModule}^n$.

### 3.2 Attention Modules

For attention, we use a simple inner-product attention that takes as input two tensors: $source$ of shape $[m, depth]$ and $target$ of shape $[n, depth]$. The attention mechanism computes the feature vector similarities at each position and re-scales according to the depth:

$$\text{Attend}(source, target) = \frac{1}{\sqrt{depth}} \cdot \text{softmax}(target \cdot source^T) \cdot source$$

To allow the attention to access positional information, we add a signal that carries it. We call this signal the *timing*, it is a tensor of any shape $[k, depth]$ defined by concatenating sine and cosine functions of different frequencies calculated upto $k$:

$$timing(t, 2d) = \sin(t/10000^{2d/depth})$$

$$timing(t, 2d + 1) = \cos(t/10000^{2d/depth})$$

Our full attention mechanism consists of adding the timing signal to the targets, performing two convolutional steps, and then attending to the source:

$$\text{attention}_1(x) = \underset{k=1,d=1}{\text{ConvStep}}(x + timing)$$

$$\text{Attention}(s,t) = \text{Attend}(s, \underset{k=4,d=1}{\text{ConvStep}}(\text{attention}_1(t)))$$

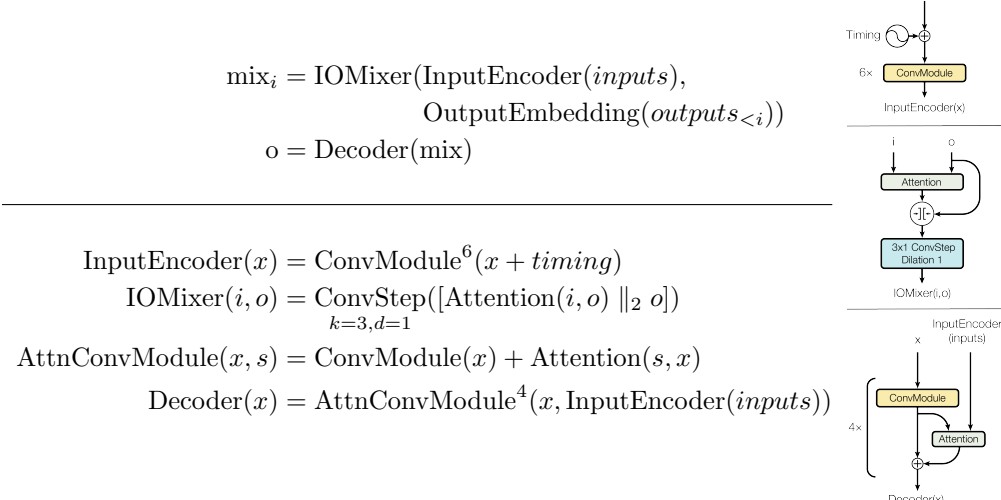

### 3.3 AUTOREGRESSIVE STRUCTURE

As previously discussed, the outputs of our model are generated in an autoregressive manner. Unlike RNNs, autoregressive sequence generation depends not only on the previously generated output, but potentially all previously generated outputs. This notion of long term dependencies has proven highly effect in NMT before. By using attention, establishing long term dependencies has been shown to significantly boost task performance of RNNs for NMT (Cho et al., 2014a). Similarly, a convolutional autoregressive generation scheme offer large receptive fields over the inputs and past outputs, capable of establishing these long term dependencies.

Below we detail the structure of the InputEncoder, IOMixer and Decoder. The OutputEmbedding simply performs a learning-embedding look-up. We denote the concatenation of tensors $a$ and $b$ along the $d^{\text{th}}$ dimension as $[a \parallel_d b]$.

$$\text{mix}_i = \text{IOMixer}(\text{InputEncoder}(inputs),$$
$$\text{OutputEmbedding}(outputs_{<i}))$$
$$\text{o} = \text{Decoder}(\text{mix})$$

$$\text{InputEncoder}(x) = \text{ConvModule}^6(x + timing)$$
$$\text{IOMixer}(i,o) = \underset{k=3,d=1}{\text{ConvStep}}([\text{Attention}(i,o) \parallel_2 o])$$
$$\text{AttnConvModule}(x,s) = \text{ConvModule}(x) + \text{Attention}(s,x)$$
$$\text{Decoder}(x) = \text{AttnConvModule}^4(x, \text{InputEncoder}(inputs))$$

## 4 RELATED WORK

Machine translation using deep neural networks achieved great success with sequence-to-sequence models (Sutskever et al., 2014; Bahdanau et al., 2014; Cho et al., 2014b) that used recurrent neural networks (RNNs) with long short-term memory (LSTM, (Hochreiter & Schmidhuber, 1997)) cells. The basic sequence-to-sequence architecture is composed of an RNN encoder which reads the source sentence one token at a time and transforms it into a fixed-sized state vector. This is followed by an RNN decoder, which generates the target sentence, one token at a time, from the state vector. While a pure sequence-to-sequence recurrent neural network can already obtain good translation results (Sutskever et al., 2014; Cho et al., 2014b), it suffers from the fact that the whole input sentence needs to be encoded into a single fixed-size vector. This clearly manifests itself in the degradation

of translation quality on longer sentences and was overcome in (Bahdanau et al., 2014) by using a neural model of attention. We use a simplified version of this neural attention mechanism in SliceNet, as introduced above.

Convolutional architectures have been used to obtain good results in word-level neural machine translation starting from (Kalchbrenner & Blunsom, 2013) and later in (Meng et al., 2015). These early models used a standard RNN on top of the convolution to generate the output. The state of this RNN has a fixed size, and in the first one the sentence representation generated by the convolutional network is also a fixed-size vector, which creates a bottleneck and hurts performance, especially on longer sentences, similarly to the limitations of RNN sequence-to-sequence models without attention (Sutskever et al., 2014; Cho et al., 2014b) discussed above.

Fully convolutional neural machine translation without this bottleneck was first achieved in (Kaiser & Bengio, 2016) and (Kalchbrenner et al., 2016). The model in (Kaiser & Bengio, 2016) (Extended Neural GPU) used a recurrent stack of gated convolutional layers, while the model in (Kalchbrenner et al., 2016) (ByteNet) did away with recursion and used left-padded convolutions in the decoder. This idea, introduced in WaveNet (van den Oord et al., 2016a), significantly improves efficiency of the model. The same technique is used in SliceNet as well, and it has been used in a number of neural translation models recently, most notably in (Gehring et al., 2017) where it is combined with an attention mechanism in a way similar to SliceNet.

Depthwise separable convolutions were first studied by Sifre (Sifre & Mallat, 2013) during a 2013 internship at Google Brain, and were first introduced in an ICLR 2014 presentation (Vanhoucke, 2014). In 2016, they were demonstrated to yield strong results on large-scale image classification in Xception (Chollet, 2016), and in 2017 they were shown to lead to small and parameter-efficient image classification models in MobileNets (Howard et al., 2017).

## 5 EXPERIMENTS

We design our experiments with the goal to answer two key questions:

- What is the performance impact of replacing convolutions in a ByteNet-like model with depthwise separable convolutions?
- What is the performance trade-off of reducing dilation while correspondingly increasing convolution window size?

In addition, we make two auxiliary experiments:

- One experiment to test the performance of an intermediate separability point in-between regular convolutions and full depthwise separability: we replace depthwise separable convolutions with grouped convolutions (sub-separable convolutions) with groups of size 16.
- One experiment to test the performance impact of our newly-introduced super-separable convolutions compared to depthwise separable convolutions.

We evaluate all models on the WMT English to German translation task and use newstest2013 evaluation set for this purpose. For two best large models, we also provide results on the standard test set, newstest2014, to compare with other works. For tokenization, we use subword units, and follow the same tokenization process as Sennrich et al. (2015). All of our experiments are implemented using the TensorFlow framework (Abadi et al., 2015). A comparison of our different models in terms of parameter count and Negative Log Perplexity as well as per-token Accuracy on our task are provided in Table 2. The parameter count (and computation cost) of the different types of convolution operations used was already presented in Table 1. Our experimental results allow us to draw the following conclusions:

- Depthwise separable convolutions are strictly superior to regular convolutions in a ByteNet-like architecture, resulting in models that are more accurate while requiring fewer parameters and being computationally cheaper to train and run.
- Using sub-separable convolutions with groups of size 16 instead of full depthwise separable convolutions results in a performance dip, which may indicate that higher separability (i.e.

| Dilations | Filter Size | Separability | Parameters (Non-Emb.) | Neg. Log | Accuracy |
|-----------|-------------|--------------|-----------------------|----------|----------|
| 1-2-4-8 | 3-3-3-3 | None | 314 M (230 M) | -1.92 | 62.41 |
| 1-2-4-8 | 3-3-3-3 | Full | 196 M (112 M) | -1.83 | 63.87 |
| 1-1-2-4 | 3-7-7-7 | Full | 197 M (113 M) | -1.80 | 64.37 |
| 1-1-1-2 | 3-7-15-15 | Full | 197 M (113 M) | -1.80 | 64.30 |
| 1-1-1-1 | 3-7-15-31 | Full | 197 M (113 M) | -1.80 | 64.36 |
| 1-2-4-8 | 3-3-3-3 | 16 Groups | 207 M (123 M) | -1.86 | 63.46 |
| 1-1-1-1 | 3-7-15-31 | Super 2/3 | 253 M (141 M) | -1.78 | 64.71 |
| 1-1-1-1 | 3-7-15-31-63 | Full (2048) | 349 M (265 M) | -1.68 | 66.71 |
| 1-1-1-1 | 3-7-15-31 | Super 2/3 (3072) | 348 M (222 M) | -1.64 | 67.27 |

Table 2: Performance on WMT EN-DE after 250k gradient descent steps.

| Model | BLEU (newstest14) | Params. |
|-------|-------------------|---------|
| SliceNet (Full, 2048) | 25.5 | 349 M |
| SliceNet (Super 2/3, 3072) | 26.1 | 348 M |
| ByteNet (Kalchbrenner et al., 2016) | 23.8 | - |
| GNMT (Wu et al., 2016) | 24.6 | 278 M |
| ConvS2S (Gehring et al., 2017) | 25.1 | - |
| GNMT+Mixture of Experts (Shazeer et al., 2017) | 26.0 | 8700 M |
| Transformer (Vaswani et al., 2017) | 28.4 | 213 M |

Table 3: Performance of our larger models compared to best published results.

groups as small as possible, tending to full depthwise separable convolutions) is preferable in this setup, this further confirming the advantages of depthwise separable convolutions.

- The need for dilation can be completely removed by using correspondingly larger convolution windows, which is made computationally tractable by the use of depthwise separable convolutions.

- The newly-introduced super-separable convolution operation seems to offer an incremental performance improvement.

Finally, we run two larger models with a design based on the conclusions drawn from our first round of experiments: a SliceNet model which uses depthwise separable convolutions and a SliceNet model which uses super-separable convolutions, with significantly higher feature depth in both cases. We achieve results that surpass all previously reported models except for the recent Transformer (Vaswani et al., 2017), as shown in Table 3, where we also include previously reported results for comparison. For getting the BLEU, we used a beam-search decoder with a beam size of $4$ and a length penalty tuned on the evaluation set (newstest2013).

## 5.1 CONCLUSIONS

In this work, we introduced a new convolutional architecture for sequence-to-sequence tasks, called SliceNet, based on the use of depthwise separable convolutions. We showed how this architecture achieves results beating not only ByteNet but also the previous best Mixture-of-Experts models while using over two times less (non-embedding) parameters and floating point operations than ByteNet.

Additionally, we have shown that filter dilation, previously thought to be a key component of successful convolutional sequence-to-sequence architectures, was not a requirement. The use of depthwise separable convolutions makes much larger convolution window sizes possible, and we found that we could achieve the best results by using larger windows instead of dilated filters. We have also introduced a new type of depthwise separable convolution, the super-separable convolution, which shows incremental performance improvements over depthwise separable convolutions.

Our work is one more point on a significant trendline started with Xception and MobileNets, that indicates that in any convolutional model, whether for 1D or 2D data, it is possible to replace convolutions with depthwise separable convolutions and obtain a model that is simultaneously cheaper to run, smaller, and performs a few percentage points better. This trend is backed by both solid theoretical foundations and strong experimental results. We expect our current work to play a significant role in affirming and accelerating this trend. We only experimented on translation, but we expect that our results will apply to other sequence-to-sequence tasks and we hope to see depthwise separable convolutions replace regular convolutions in more and more use cases in the future.

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
