# OpenReview forum: "Depthwise Separable Convolutions for Neural Machine Translation"
_ICLR.cc/2018/Conference — Accept (Poster)_

### Official Review · AnonReviewer2 · 2017-11-25
**More experiments**

**Rating:** 5
**Confidence:** 4

**Review:**

Pros:
- new module
- good performances (not state-of-the-art)
Cons:
- additional experiments

The paper is well motivated, and is purely experimental and proposes a new architecture. However, I believe that more experiments should be performed and the explanations could be more concise.

The section 3 is difficult to read because the notations of the different formula are a little bit heavy. They were nicely summarised on the Figure 1: each of the formula' block could be replaced by a figure, which would make this section faster to read and understand.

I would have enjoyed a parameter comparison in Table 3 as it is claimed this architecture has less parameters and additional experiments would be welcome. As it does not reach the state-of-the-art, "super separable convolutions" could be compared on other tasks?

minor:
"In contrast, regular convolutional layers break
this creed by learning filters that must simultaneously perform the extraction of spatial features and
their merger into channel dimensions; an inefficient and ineffective use of parameters." - a verb is missing?

---

> ### Comment · AnonReviewer2 · 2017-12-08
> **Encouraging the authors**
>
> I think there is a kind of consensus about the reviews of this paper. I would like to kindly encourage the authors to modify the sections 2&3, in order to incorporate the changes we requested, and to give some additional numerical results with a little bit more comments. In this case and if the modifications are relevant, I would be happy to raise my rating.

---

> ### Author Response · Authors · 2018-01-03
> **Thank you for your help, new revision addresses the issues**
>
> We are very grateful for helping us improve the paper.
>
> The reviewer wrote: "section 3 is difficult to read because the notations of the different formula are a little bit heavy. They were nicely summarised on the Figure 1: each of the formula' block could be replaced by a figure, which would make this section faster to read and understand." We took this very seriously and we have re-arranged the whole presentation of equations in Section 3. In the new revision, every set of equations comes together with the corresponding Figure, and the figures were slightly re-drawn to match the equations closer. We hope that this addressed the main concern about presentation.
>
> As for Table 3, we found it hard to get the parameter count for every one of the earlier models presented in the table. But we asked the authors of the other papers and we will try to add one more revision with parameter counts.
>
> As for not reaching state-of-the-art, we believe that it is due to the lack of self-attention in the decoder. Please see the comment to Review 1 above where we discuss this.

---

### Official Review · AnonReviewer3 · 2017-11-27

**Rating:** 7
**Confidence:** 4

**Review:**

The paper proposes to use depthwise separable convolution layers in a fully convolutional neural machine translation model. The authors also introduce a new "super-separable" convolution layer, which further reduces the computational cost of depthwise separable convolutions. Results are presented on the WMT English to German translation task, where the method is shown to perform second-best behind the Transformer model.

The paper's greatest strength is in my opinion the quality of its exposition of the proposed method. The relationship between spatial convolutions, pointwise convolutions, depthwise convolutions, depthwise separable convolutions, grouped convolutions, and super-separable convolutions is explained very clearly, and the authors properly introduce each model component.

Perhaps as a consequence of this, the experimental section feels squeezed in comparison. Quantitative results are presented in two fairly dense tables (especially Table 2) which, although parsable after reading the paper carefully, could benefit from a little bit more information on how they should be read. The conclusions that are drawn in the text are stated without citing metrics or architectural configurations, leaving it up to the reader to connect the conclusions to the table contents.

Overall, I feel that the results presented make a compelling case both for the effectiveness of depthwise separable convolutions and larger convolution windows, as well as the overall performance achievable by such an architecture. I think the paper constitutes a good contribution, and adjustments to the experimental section could make it a great contribution.

---

### Official Review · AnonReviewer1 · 2017-11-27

**Rating:** 7
**Confidence:** 3

**Review:**

This paper presents the SliceNet architecture, an sequence-to-sequence model based on super-dilated convolutions, which allow to reduce the computational cost of the model compared to standard convolution. The proposed model is then evaluated on machine translation and yields competitive performance compared to state-of-the-art approaches.

In terms of clarity, the paper is overall easy to follow, however I am a bit confused by Section 2 about what is related work and what is a novel contribution, although the section is called “Our Contribution”. For instance, it seems that the separable convolution presented in Section 2.1 were introduced by (Chollet, 2016) and are not part of the contribution of this paper. The authors should thus clarify the contributions of the paper.

In terms of significance, the SliceNet architecture is interesting and is a solid contribution for reducing computation cost of sequence-to-sequence models. The experiments on NMT are convincing and gives interesting insights, although I would like to see some pointers about why in Table 3 the Transformer approach (Vaswani et al. 2017) outperforms SliceNet.

I wonder if the proposed approach could be applied to other sequence-to-sequence tasks in NLP or even in speech recognition ?

Minor comment:
* The equations are not easy to follow, they should be numbered. The three equations just before Section 2.2 should also be adapted as they seem redundant with Table 1.

---

> ### Author Response · Authors · 2018-01-03
> **Grateful for the review, updated equations in the new revision, addressing Transformer results.**
>
> We are very grateful for the review.
>
> As for the suggestion to improve presentation and equations, we have uploaded a new revision with diagrams put together with equations in a new way (inspired by another review). We hope this makes it easier to understand.
>
> As for this point: " I would like to see some pointers about why in Table 3 the Transformer approach (Vaswani et al. 2017) outperforms SliceNet." -- let us explain how Transformer has crucial architectural parts missing from SliceNet. One key part of Transformer is self-attention in the decoder: an attention layer that allows the decoder to attend to previously generated (decoded) words. This is a main innovation of the Transformer architecture and it is missing from SliceNet (as we started working on SliceNet before the Transformer paper). We only have the encoder-decoder attention known from previous sequence-to-sequence models (i.e., the decoder can attend to the encoder, but not to previously decoded words). We believe that this difference is responsible for the difference in results: as far as we know, no architecture without self-attention in the decoder has shown better results than SliceNet. It should also be possible to combine the decoder self-attention with SliceNet -- some results (with non-separable convolutions) are already coming up as SNAIL for image generation (https://arxiv.org/abs/1712.09763). We believe that the techniques we present in this paper can be used to extend SNAIL to get additional improvements, and it should also work for tasks like image generation, parsing, summarization and others.

---

### Decision · Program_Chairs · 2018-01-29
**ICLR 2018 Conference Acceptance Decision**

**Decision:**

Accept (Poster)

**Comment:**

Paper explore depth-wise separable convolutions for sequence to sequence models with convolutions encoders.
R1 and R3 liked the paper and the results. R3 thought the presentation of the convolutional space was nice, but the experiments were hurried. Other reviewers thought the paper as a whole had dense parts and need cleaning up, but the authors seem to have only done this partially.
From the reviewers comments, I'm giving this a borderline accept. I would have been feeling much more comfortable with the decision if the authors had incorporated the reviewers' suggestions more thoroughly..